# Integration of Remote Sensing and GIS to Extract Plantation Rows from A Drone-Based Image Point Cloud Digital Surface Model

**Nadeem Fareed** [1,*]  **and Khushbakht Rehman** [2]

1    Department of Geomatics, National Cheng Kung University, No.1, University Road, Tainan 701, Taiwan
2    Institute of Geology, University of Azad Jammu and Kashmir, Muzaffarabad 46000, Pakistan;
     khushbakhtrehman280@gmail.com
*    Correspondence: nadeem.esci@gmail.com; Tel.: +886-963-534-464

**Abstract:** Automated feature extraction from drone-based image point clouds (DIPC) is of paramount importance in precision agriculture (PA). PA is blessed with mechanized row seedlings to attain maximum yield and best management practices. Therefore, automated plantation rows extraction is essential in crop harvesting, pest management, and plant grow-rate predictions. Most of the existing research is consists on red, green, and blue (RGB) image-based solutions to extract plantation rows with the minimal background noise of test study sites. DIPC-based DSM row extraction solutions have not been tested frequently. In this research work, an automated method is designed to extract plantation row from DIPC-based DSM. The chosen plantation compartments have three different levels of background noise in UAVs images, therefore, methodology was tested under different background noises. The extraction results were quantified in terms of completeness, correctness, quality, and F1-score values. The case study revealed the potential of DIPC-based solution to extraction the plantation rows with an F1-score value of 0.94 for a plantation compartment with minimal background noises, 0.91 value for a highly noised compartment, and 0.85 for a compartment where DIPC was compromised. The evaluation suggests that DSM-based solutions are robust as compared to RGB image-based solutions to extract plantation-rows. Additionally, DSM-based solutions can be further extended to assess the plantation rows surface deformation caused by humans and machines and state-of-the-art is redefined.

**Keywords:** UAVs; point clouds; PA; GIS; remote sensing; crop row detection

## 1. Introduction

Row plantation is a common practice in conventional horizontal farming or gardening systems. The system consists of plantation in a linear pattern in one direction and usually, east–west orientation. A uniform seedling spacing and parallel equidistant lines are maintained to maximize light exposure, enhance maximum yields, and inventory convenience. Row plantation is in practice in most crops of direct-seeded, transplanted, or grown from vegetative or dormant sprouts [1–3]. Precision agriculture (PA) is blessed by row plantation not only to maximize the crop yield but also to bring advanced and cost-effective technology and methods to ease the crop inventory and management practices by utilizing the global navigation satellite systems (GNSS), imaging and scanning sensors, and geographic information system (GIS) [4–7].

Drones, or commonly known as unmanned aerial vehicles (UAVs), have shown promising advancement in the development of a low-cost platform and imaging sensors during the last two decades [8,9]. In PA, the use of UAVs systems is a newly developed, robust, and an emerging fine-scale

remote sensing method that has numerous advantage over space-borne and field-based traditional practices [10]. The UAVs are flown with an unprecedented spatio-temporal resolution to acquire very-high-resolution (VHR) images in optical, infrared, and thermal regions of the electromagnetic spectrum [11,12]. A UAV is flexible to fly at any desired date and time without being limited by cloud cover, ground conditions unlike human operators, and ground-based systems [13]. Furthermore, the use of active remote sensing technology such as light detection and ranging (LiDAR) has shown exceptional growth in remote sensing technology, and wide-area coverage to provide high quality and the most accurate 3D point clouds of forests and farms to facilitate the PA [14,15]. However, LiDAR is still considered an expensive technology for middle and small-scale farmers, therefore, a low-cost solution has emerged in the form of UAVs passive imaging sensors [16].

On the other hand, along with UAVs system development, digital photogrammetry techniques such as structure from motion (SfM) have significantly provided a mature and low-cost solution to construct drone-based images point clouds (DIPC) and orthophoto from UAVs overlapping VHR images [17,18]. In order to construct a DIPC, SfM is based on the principle of different viewing angles of the same scene to identify well-defined geometric features. The SfM workflow is the automation of incremental construction of the 3D shape of an object or scene through *Feature Extraction*, *Feature Matching*, and *Geometric Verification.* The most prominent objects are extracted from the overlapping input images in the *Feature Extraction* phase. The extracted objects are known as *Key Points* [19]. This is further followed by the *Feature Matching* between extracted *key points* from two different but overlapping images. Finally, the geometric verification is carried out to fix the outliers in the *Feature Matching* step. The SfM pipeline consists of a large variety of algorithms to construct 3D point clouds [20]. In an agricultural environment, the SfM is well documented in previous studies [21–24]. A DIPC is frequently used to extract digital elevation models (DEMs), digital surface models (DSM) [22,25,26], canopy height models (CHMs) [27–29], and crop surface models (CSMs) [30] at user-defined cell-resolution through automation. A large number of studies have already demonstrated the feasibility of using DIPC to estimate the biophysical properties of tree strands and forest structure attributes such as tree-crown with reasonable accuracy [31].

In order to satisfy the rising demand for sustainable farming and PA, autonomous solutions to extract features from UAVs images in a timely and cost-effective manner are essential [32]. A number of studies have already been conducted to extract plantation rows from UAVs imagery. The identification of plantation rows in UAVs images, and thereby its automatic extraction, is of paramount importance for crop planning in terms of correction of failures in sowing, early-growth-stage monitoring, seedling counts, yield-estimation, and crop harvesting [33]. For the detection and extraction of plantation rows from UAVs images, a number of different methods have been implemented in the past. Most of these methods are image-based solutions, in which RGB or multispectral images were transformed into binary images of vegetative and non-vegetative objects through color indices, image segmentation techniques of supervised and unsupervised clustering [34–37], image transformation techniques such as RGB to HSV (Hue, Saturation, Value) [8], and principal component analysis (PCA) [8]. This is followed by a number of line detection algorithms to separate plantation rows from background soil and weeds. The early attempts were made to use Hough transform, Hough transforms with least square [38,39], Hough transforms on tiled images of a curved plantation [33], and fast Fourier transform (FFT) [40,41]. The recent advances in object-based image analysis (OBIA) to detect linear objects based on their geometry, and convolutional neural networks (CNN) have also shown very promising results [37,42,43]. On the contrary, robotic solutions in real-time crop row detection has been investigated by Winterhalter et al. (2019), in which laser range scanners with a vision camera were mounted on a robot. The number of crop row detection methods based on elevation maps are rare [2].

The existing research has significant gaps in terms of extraction of plantation rows from UAV-based imaging solutions. A large number of studies are based on test images of small study-sites without a detailed description of area coverage and terrain morphology. Additional parameters such as row orientation angle and UAV flying trajectory information were also required [8,44]. Furthermore, these studies were conducted in a comparatively cleaner environment where plantations are uniformly

distributed with minimum background noise of soil classes, weeds, and dead plantation of previous crops [45]. Some of the studies were missing the accuracy assessment of the extracted results [8]. In addition to this, the proposed methods were based on complex mathematical algorithms usually not available within frequently used remote sensing, and GIS software packages [44]. Finally, most of these methods have been implemented and tested for UAV-based multispectral and RGB-images only where elevation-based solutions were rare. On the other hand, UAV-based orthophotos and DIPC are the most common products and frequently produced from the UAV image data [46–48]. In order to utilize the 3D point clouds, a vast variety of software packages are available in proprietary and open source platform domains [49,50]. In terms of plantations-row detection and, therefore automated extraction, the 3D point clouds have been hardly investigated and utilized in academia and industry.

In PA, the extraction of plantation rows is very challenging at early-growth-stage using RGB-imagery only when plant strands are significantly smaller. This problem is further compounded by the presence of background noise of weeds, loams, and dead plantations. Therefore, RGB-image based solutions have difficultly extracting plantation-rows. Therefore, the objective of the proposed study is (a) to utilize a DIPC-based DSM to extract the plantation rows by exploiting the elevation only, consequently, the proposed solution is also applicable and extendable for LiDAR-based DSM; (b) the use of a DIPC-based solution to identify the damaged and compromised plantation rows in order to assess the damage in plantation rows to warrant the plantation-row restoration for optimized yield to ensure the PA; (c) finally, to evaluate the potential of some frequently used remote sensing and GIS software packages to automate the plantation-row extraction by utilizing the inherently available GIS and image processing tools.

The rest of this article consists of three sections and their respective subsections. Section 2 details a detail description of the study sites, data acquisition, and processing of DIPC into the ground and non-ground features along with automated point cloud classification issues. This is followed by the proposed methodology to extract plantation-rows. Section 3 is devoted to explaining the plantation-row damage assessment and plantation-row extraction results along with accuracy assessment. The subsections quantify the results of the proposed research to compare and contrast with existing research. Section 4 concludes the research highlighting the significant findings and merits over existing methods.

## 2. Materials and Methods

### 2.1. Study Area

The study area is located in the southeastern part of Victoria, Australia. The field plot of 2 km long and 0.5 km wide centered at coordinates (142°59′5.603″ E, 38°17′52.095″ S) was selected for the proposed methodology. The study site consists of three plantation compartments of sizes 44, 21, and 17 ha, respectively (Figure 1a). The seedlings rows are oriented north–south for compartment-A and row orientation is east–west for compartment-B and compartment-C. The overall spectral variability of the study area is dominated by soil (Figure 1b) and background noise of shrubs, wild species of woody weeds, and dead plantation of the previous crop (Figure 1b–d). *Eucalyptus globulus* (*E. globulus)* were planted in June 2019, therefore, the plants were 6 months old (Figure 1d). The original plantation has achieved 1000-stems per hectare with 2.2 m seedling spacing apart with an inter-row spacing of 4.5 m.

Generally, the tree plantations are well spaced with minimal background noise, however, this is always not the case for each plantation site. The chosen site has a significant background noise of wild species of different spectral profiles, a number of classes of dry and wet soil, and remains of dead plants. Furthermore, *E.globulus* plantation is significantly smaller in size (Figure 1b,d) and overlaps with wild species. Three different compartments (A, B, and C) also possess three different levels of background noises. The compartment-A has significant dead plantation background noise dominating the soil and wild species (Figure 2a), the compartment-B is comparatively clean in terms of the background noise of wild species and dead plantation (Figure 2b), and compartment-C is dominated by grass, shrubs, and wild species (Figure 2c). Therefore, three different levels of background noises were chosen to test the proposed methodology.

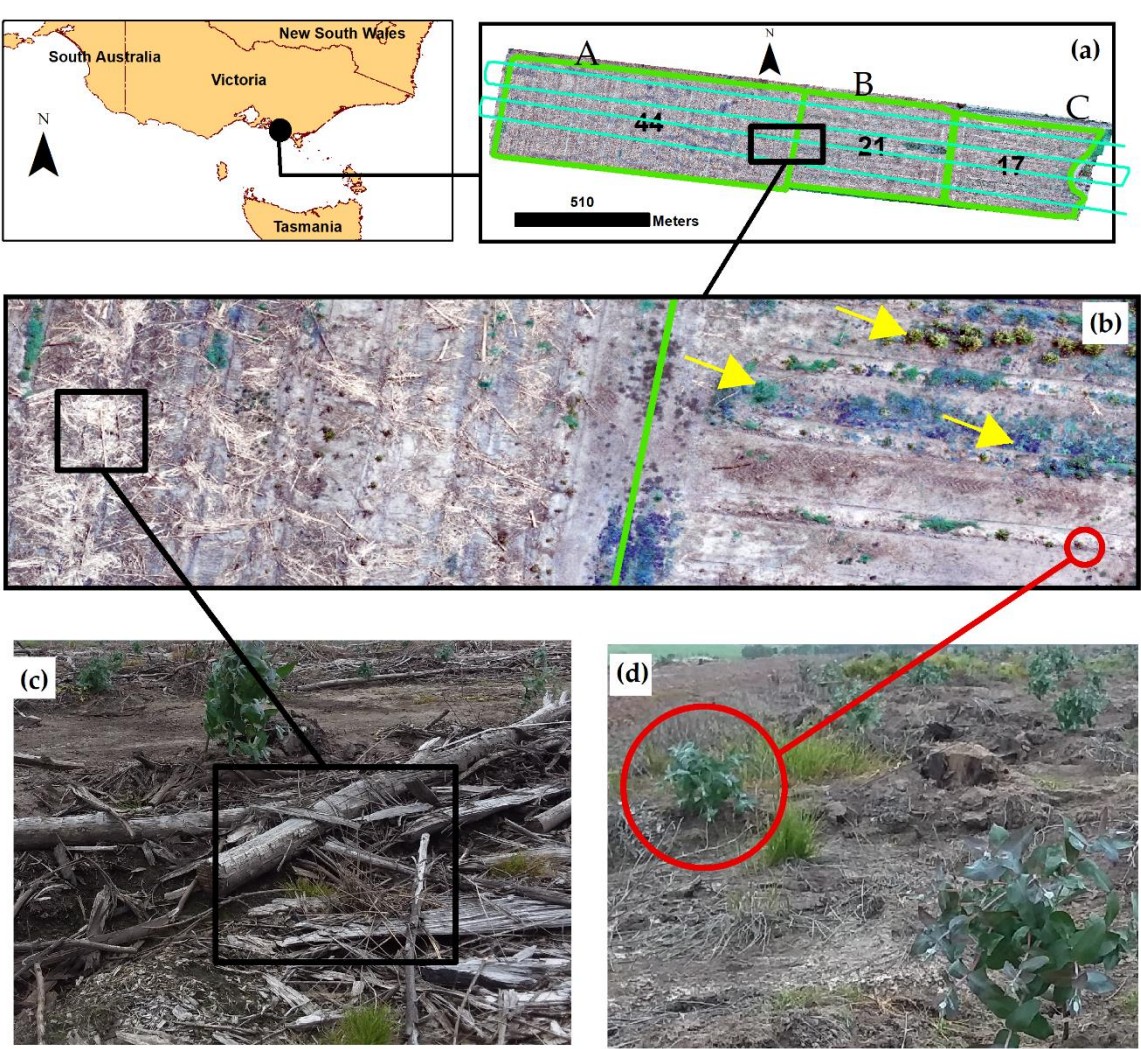

**Figure 1.** Study area map: the base map shows the location of the study area. (**a**) The inset map shows the UAVs mapped plantation area for three consecutive compartments; (**b**) the plantation at a 20,000-scale, where plantation at regular intervals can be seen along with background soil and background noise of weeds indicated by yellow colored arrows; (**c**) remains of dead plantation or previous crop; (**d**) *E. globulus* plantation along with background noise of grass.

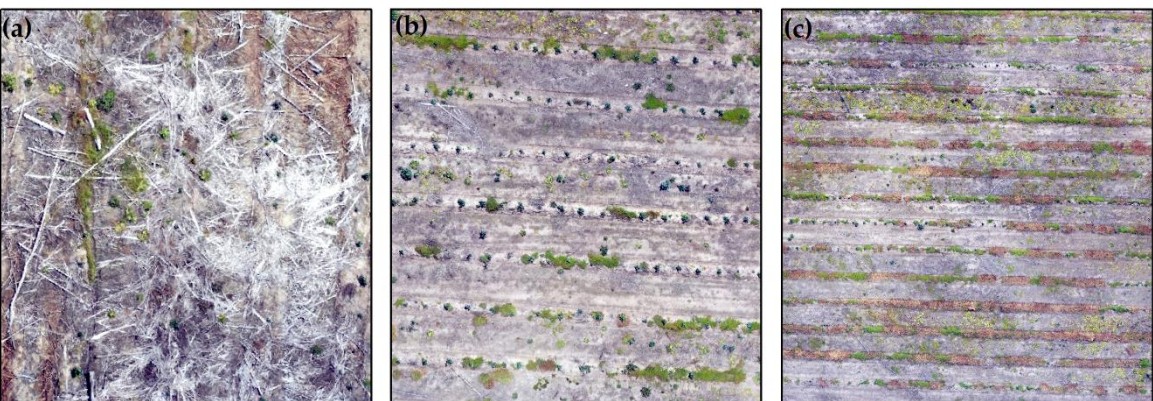

**Figure 2.** The level of noise in three different plantation compartments: (**a**) compartment-A is dominated by dead plant background noise; (**b**) compartment-B is fairly clean from dead plants and wild species dominating by soil; (**c**) and compartment-C is subject to the background noise of wild species.

### 2.2. UAV Image Acquisition and Processing

The UAVs was flown to image the proposed study site using commercial DJI Mavic-2 pro during the clear sky condition on December 11, 2019, at an elevation of 50 m above the ground. The flight trajectory was oriented east–west as shown in Figure 1a. The UAV was equipped with the Hasselblad-L1D-20c 1-inch complementary metal-oxide-semiconductor (CMOS) sensor to acquire Nadir-RGB images at 20MP with a 70% horizontal and lateral overlap. The Hasselblad L1D-20c is state-of-the-art for aerial imaging by offering improved lowlight imaging capabilities. Furthermore, with Hasselblad Natural Color Solution (HNCS), images are captured with optimal color accuracy and the use of the sensor is well-documented for aerial imaging [51,52].

The acquired raw images were uploaded to Agisoft PhotoScan Professional Edition 1.2.6 (Agisoft, St. Petersburg, Russia) for preprocessing and processing to obtain a DIPC and Orthophoto. The Agisoft PhotoScan is a combination of proprietary algorithms with a user-friendly workflow and general user interface (GUI). The workflow was automated through batch processing. Batch processing is a graphical interface where image processing tasks can be automated by listing each task in a hierarchy. The automated workflow is shown in Figure 3.

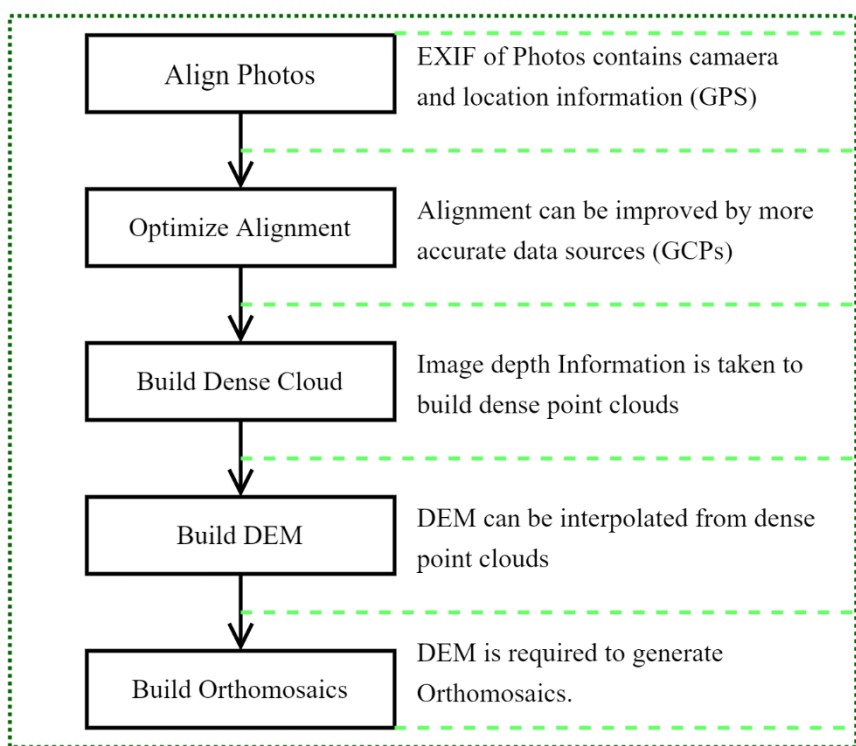

**Figure 3.** The automated workflow implemented in Agisoft photoscan batch-processing. A batch process accomplishes each task automatically and saves each task on the disk before moving to the next task in the chain.

The workflow implementation in Agisoft is user-friendly as each task can be automated. However, before processing the data in Agisoft there are certain quality checks in order to get the desired accuracy and quality of the end-products of DSM, CSM, CHM, and DEM. SfM algorithms derive 3D construction of scenes from acquired photos. Therefore, two parameters are of utmost importance: first, the position of the cameras in space; and second, the quality of acquired images. Once the photos are uploaded in Agisoft, the software determines the position of each camera by reading the Exchangeable Image File Format (EXIF) of an image. The EXIF file contains all the information about imaging sensor and location coordinates (x,y) along with elevation (z) if the sensor was equipped with an onboard GPS unit (Table 1). The software positions the camera in space solely on the basis of EXIF information

contained in each image. A consumer grade GPS unit can result in an error of a few meters with respect to the actual position of the acquired images [53] as RMSE of onboard GPS is shown in Table 1. In order to obtain an accuracy of a few centimeters, highly accurate ground control points (GCPs) can be placed in the desired study area. The accurate position of GCPs is measured by the GNSS survey in real-time-kinematic (RTK) mode [54,55]. Therefore, the position of each camera can be optimized in an optimized alignment phase in Agisoft (Figure 3). The optimization in cm is highly desired for some applications of PA such as crop surface monitoring by using crop surface models (CSMs) where absolute height is of paramount importance [30]. For other applications, such as extracting plantation rows from a DSM in our case, the consumer grade GPS has proven sufficient as plantation rows were 2 m wide and 4.5 m apart with regular spacing in flat terrain. However, for other study sites, the terrain morphology can vary greatly, therefore, the accuracy requirement should be well anticipated before the acquisition of UAV imagery. A study by Forlani et al. (2018) found DSMs quality assessment is worth considering in order to choose a UAV system to acquire data. The second important aspect of SfM-based dense point cloud construction is image quality. A number of factors can affect image quality such as image acquisition time, weather conditions, lighting conditions, and imaging-sensor being used [30]. An imaging-sensor should be calibrated through camera calibration in order to assess the camera lens and hardware performance [56,57]. Agisoft Photoscan automatically checks the image quality, therefore, deformed and blurred images can be screened out before processing a DIPC. In this study, out of 1459 images, seven images were tagged as bad quality and therefore removed from the software before the image alignment phase. These factors can be planned way before the UAV flight. However, there are some other factors that are important during flight and data acquisition time such as overlaps between consecutive images and sensor-height from the ground. The detail explanation of image quality and thereby extraction of the dense point cloud is beyond the scope of this study, therefore for a deep understanding of underlying concepts, techniques, and robust methods, a study by Turner et al. (2012) can be consulted [23]. The parameters of the camera before and after alignment optimization are shown in Table 2.

**Table 1.** A detail description of acquired images and DIPC.

| Item | Description |
| --- | --- |
| Imaging sensor | Hasselblad-L1D-20c |
| Shutter speed | 1/1000 |
| Image count | 1452/1459 |
| Orthomosaic image density | 4 images/pixel |
| Image resolution | $5472 \times 3648$ ($\approx$20MP) |
| GSD orthomosaic | 1.07cm/px |
| DIPC points | 2,80,84,916 |
| Point spacing | 0.2 m |
| UAV onboard GPS RMSE | 1.23 m |

**Table 2.** Parameters of camera alignment before and after optimization.

| | Flying Altitude | Calculation Quality | Tie-Points |
| --- | --- | --- | --- |
| Before Optimization | 48.2 m | High | 471,592 |
| After Optimization | 49.9 m | High | 490,160 |

Before optimized alignment, the onboard GPS coordinates were used to find the position of camera locations. However, a consumer grade GPS has a positional error of a few meters as shown in Table 1. Therefore, camera alignment was not perfect. The camera offset from its true location can offset the images by a few meters thereby resulting in less overlap between two consecutive images. This is the reason that fewer tie-points were extracted from overlapping images before the optimization process (Table 2). However, the camera interior and exterior parameters can be used during the optimal

alignment and camera locations were readjusted which results in a better approximation of camera locations as elevation was adjusted to 49.9 m (close to the actual flying height of 50 m) as compared to 48.2 m before optimization. This can significantly increase the overlap between two consecutive images and results in more tie-points as shown in Table 2. Even optimized alignment by using the camera parameters still has some degree of error in alignment. However, the accurate GCPs obtained through an RTK can further increase the alignment accuracy thereby resulting in the desired accuracy in cm.

After the optimized alignment, tie-points were used to construct dense point cloud and calculation quality was set as high. The computation of dense point cloud uses different families of algorithms such as MVS algorithms or pairwise depth map construction. Images depth was exploited by a well-known approach of scale-invariant feature transformation (SIFT) algorithm, then the internal and external camera orientation parameter was estimated by a greedy algorithm. Finally, the bundle-adjustment algorithm was used to fine tune the DIPC [30].

In order to access the DIPC area-wise point density, a grid of 30 m cell size was created for three compartments. The total number of points was calculated for each grid cell of three compartments. The area-wise point density distribution is shown in Figure 4. Compartment-A showed a uniform point density distribution ranging from 36,440 to 37,371 points for each grid cell (Figure 4a). The Compartment-B point density distribution was similar to compartment-A, except for a few locations of water bodies (ponds) and highly vegetated areas of trees and weeds (Figure 4b). On the contrary, the point density for compartment-C showed a decreasing point count trend from north–west to south–east (Figure 4c). Similarly, the orthophoto visual interpretation also indicates a similar trend of wild species of weed distribution for compartment-C. Therefore, DIPC construction was affected by weed species. The failure can be attributed to the uniform texture of weed plants providing less matching points to construct a dense point cloud by photogrammetry software.

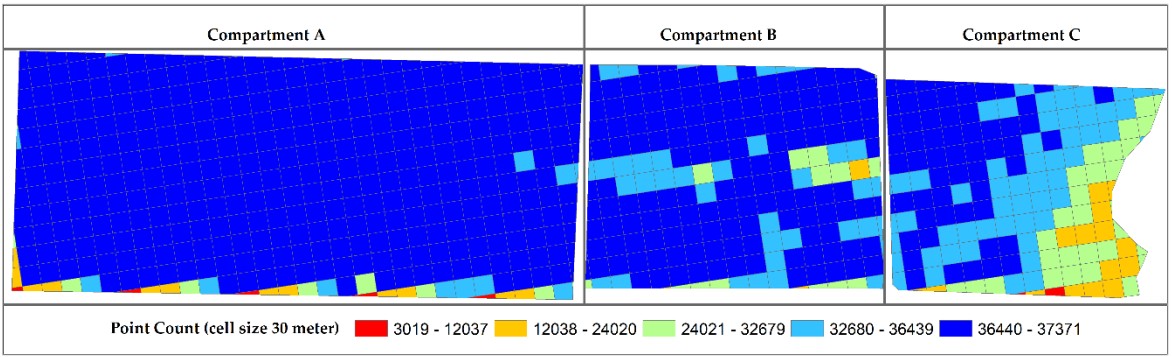

**Figure 4.** DIPC point density for three compartments: (**a**) a uniform point density for compartment-A; (**b**) a uniform point count distribution for compartment-B, except water bodies and vegetated areas in the center of the site; (**c**) point distribution is highly affected by wild species in the south-eastern part of compartment-C.

A DIPC generated through automated photogrammetric techniques is a raw point cloud. A raw point cloud does not have the information about features it represents, but each point carries the location information (x, y) along with its respective elevation (z) value (Figure 5a-1, b-1, c-1 of profiles (a), (b), and (c)). In order to classify a raw point cloud into distinct features or objects such as ground (terrain) or non-ground points (trees and buildings), a number of point cloud classification algorithms exist to automate the process [58–60]. The automated raw point cloud classification has a varying degree of classification accuracy and manual classification is subject to a significant amount of time, therefore manual point cloud classification was not carried out. The DIPC was automatically classified into the ground and non-ground points by using the conservative ground point classification algorithm available in ArcGIS Desktop (10.7). The said algorithm has a tighter restriction on ground slope variation to separate the ground points from non-ground points of low-laying vegetation such as grass

and weeds. The classification results are shown in Figure 5 (d-1, e-1, and f-1) for profiles (d), (e), and (f). In comparison with raw point cloud as depicted in Figure 5 (a-1, b-1, and c-1), the automated extraction of ground points from a raw point cloud is subject to a significant loss in row information, especially the row tops. Therefore, the DSM created from classified ground returns is noisier as three respective profiles for ground returns are shown in Figure 5d-1, e-1, and f-1; green, red, and purple profiles. The constructed profile for three different locations has indicated that more smoothing was required to denoise the respective DSM, thereby increasing the processing time. On the other hand, surface profiles extracted from raw point cloud are smoother where row shapes were preserved as compared to the background as shown in Figure 5 (a-1, b-1, and c-1). This evaluation suggests that a raw point cloud has sufficient information to create a DSM without loss of available height information. Due to this reason, the proposed methodology was based on a DSM from a raw point cloud. DSM generated from a raw point cloud has already been evaluated by many authors [61,62]. It is clear that an unclassified DIPC has sufficient information about plantation rows Figure 5 (a-1, b-1, and c-1). The elevated plantation rows can be distinguished clearly from the background noise of soil and weeds. Therefore, a raw point cloud was a preferred choice to create a high-resolution DSM of cell size 0.2 m. The entire workflow to extract plantation rows from DIPC-based DSM is shown in Figure 6.

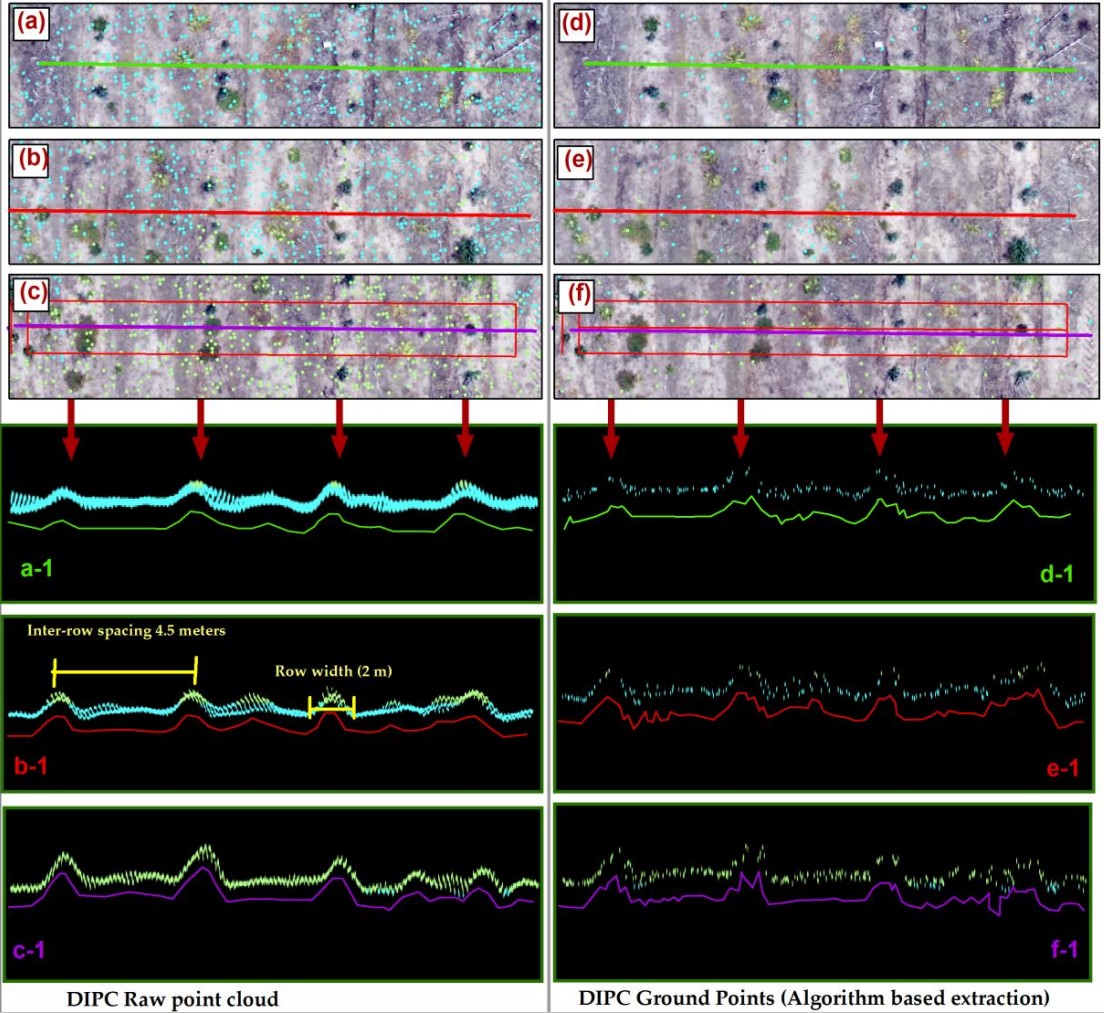

**Figure 5.** A raw DIPC vs. ground points extracted from DIPC. (**a–c**) Profiles of raw DIPC. a-1, b-1, and c-1 are 3D profiles of raw-DIPC and corresponding solid-lines of DSM surface profiles. (**d–f**) Profiles of ground returns. d-1, e-1, and f-1 are 3D profiles of DIPC ground returns and corresponding solid-lines of DSM surface profiles. The profiles were slightly offset to show respective point clouds.

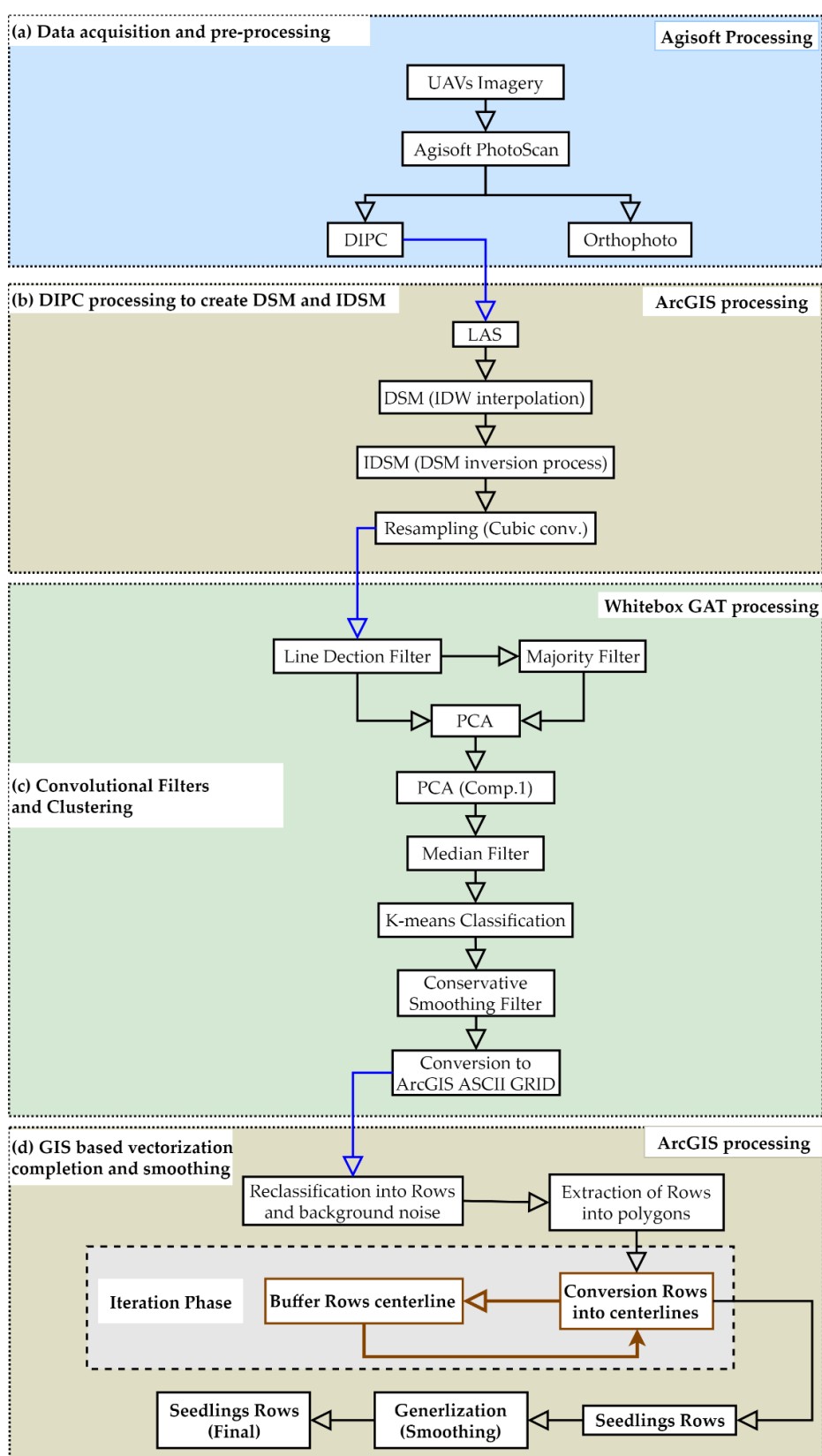

**Figure 6.** UAV data processing workflow: (a) UAV data processing to obtain DIPC and Orthomosaic (Figure 3); (**b**) raw-DIPC processing to create DSM and IDSM; (**c**) convolutional filtering, image transformation, unsupervised clustering; (d) GIS-based vectorization and processing to obtain plantation rows.

### 2.3. DIPC Processing to Create DSM

The DIPC was exported into the American Society for Photogrammetry and Remote Sensing (ASPRS) LAS-file format from Agisoft. DIPC was further processed in ArcGIS Desktop.10.7 (Environmental System Research Institute, Redlands, CA, 2019). The LAS file was interpolated by using inversed distance weightage (IDW) to obtain the digital surface model (DSM) of the study area at a cell resolution of 0.2 m [63]. The high cell resolution was chosen to capture a very detailed local topography. The farm topography was dominated by two parameters: first, the elevated plantation-rows; and secondly, the flat terrain between two consecutive plantation rows (Figure 7a). In order to automate the plantation rows delineation, the original DSM was inverted by using the following formula.

$$DSM \ (Inversion) = ((DSM - Max) * -1) + Min \tag{1}$$

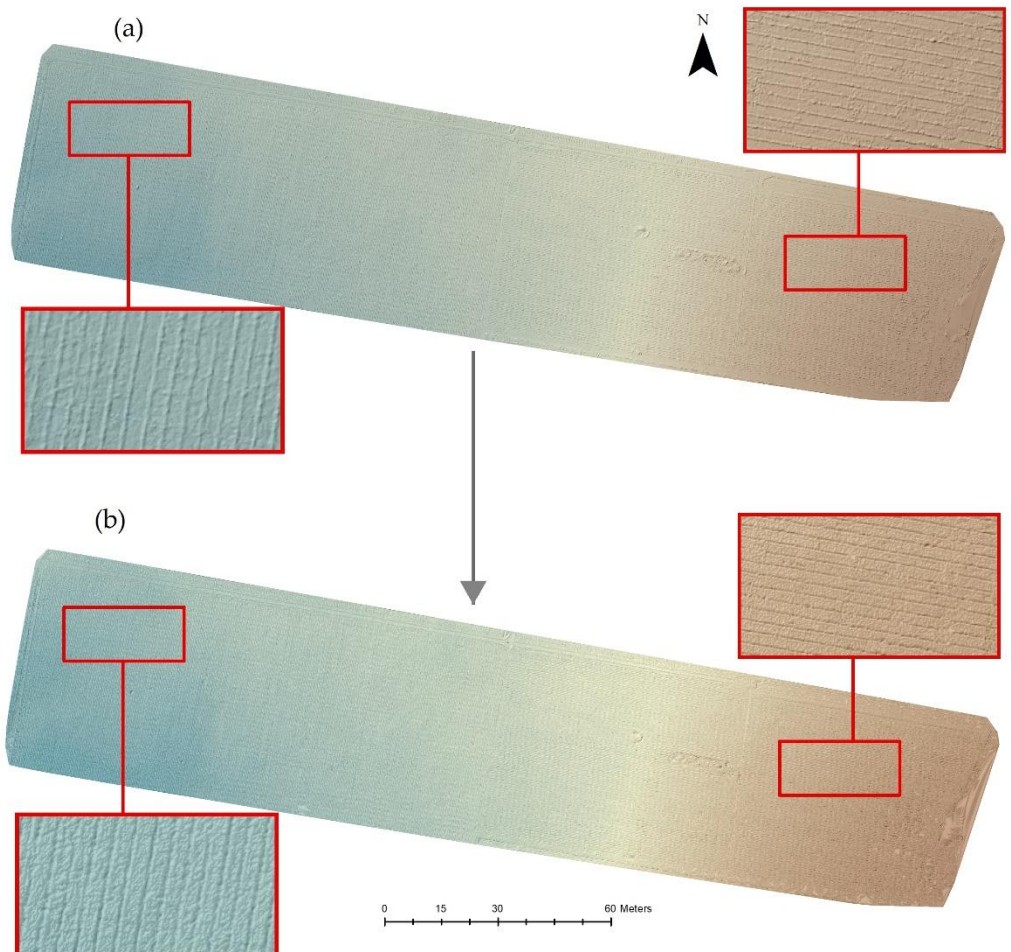

**Figure 7.** The process of DSM inversion. (**a**) DSM obtained after interpolating raw-DIPC; (**b**) IDSM obtained after inverting DSM. The plantation lines become more prominent in IDSM.

In the DSM, the plantation rows were elevated as compared to surroundings terrain, in order to emphasize the line shape of the plantation-rows, the original DSM was inverted as shown in Figure 7b.

The IDSM was further resampled to 0.4 m to increase the processing speed. In addition to this, the Cubic convolutional resampling technique was used to resample the IDSM. Cubic convolution interpolates the value of each pixel by fitting a smooth curve through 16 neighboring pixels. Therefore, the IDSM obtained after resampling is smoother and background noise of pits and image roughness were minimized as compared to 0.2 m original IDSM. However, the plantation rows information

was not affected by this process as depicted in Figure 8. This should be noted that the DSM-based plantation-row extraction entirely depends on row width, depth, and elevation differences as compared to inter-row spacing. If data acquisition is capable of capturing all the said parameters, the extracted results will not be jeopardized.

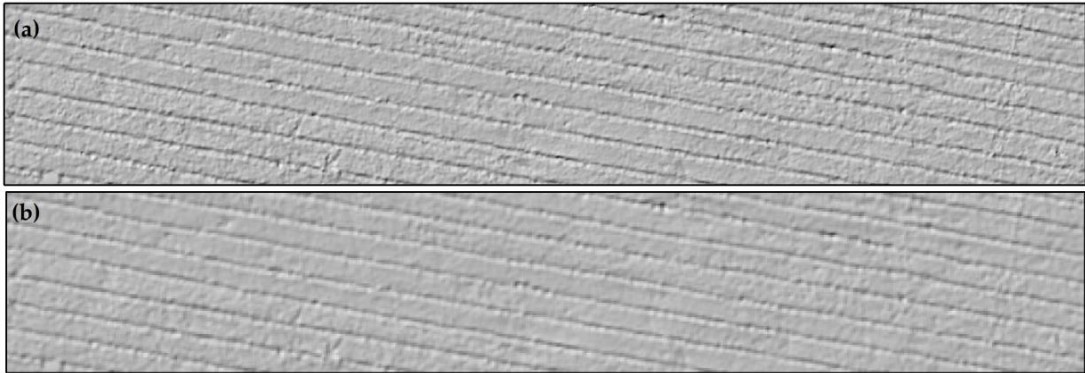

**Figure 8.** The effect of cubic convolution resampling. (**a**) IDSM before resampling; (**b**) the image becomes smoother and background noises were minimized after sampling.

### 2.4. Convolutional Filtering Image Transformation and Unsupervised Classification

The IDSM produced in the preprocessing step was further exported into Whitebox GAT Montreal (v.3.4.0) format. Whitebox GAT is a free and open source software (FOSS) remote sensing and GIS package. The software package contains more than 360 plug-ins and tools to carry out geospatial analysis for raster and vector data format. In the Whitebox GAT environment, spatial filtering techniques were used to extract the plantation rows from IDSM. Spatial filters are based on the idea of moving window or kernel of a specified sized. The kernel or convolution process performs a specified mathematical manipulation for noise reduction, smoothing, sharpening, edge detection, line detection, skeletonization, line thinning, and mathematical morphology on a raster image [64]. In order to extract the plantation rows from IDSM, line-detection filter (LDF) was used to detect the line features from IDSM. The tool has four (3*3) convolutional-based filters to detect lines in horizontal, vertical, 135°, and 45° orientation in a raster image. In the proposed study area, rows were oriented in east–west (horizontal) or north–south (vertical) direction, therefore, LDF-horizontal, and LDF-vertical were used to detect the lines in IDSM. The detection results are shown in Figure 9. However, the plantation-row orientation is not exactly horizontal, and a vertical but LDF-horizontal and LDF-vertical performance was not affected by minor differences in line orientation. Therefore, for the plantation sites which do not exactly meet the criteria of horizontal and vertical orientation, the two other orientation filtering schemes of 45° and 135° can be applied.

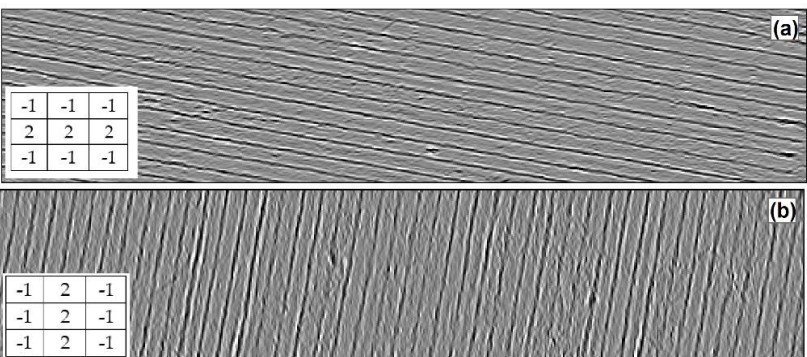

**Figure 9.** Line detection filters to detect lines in IDSM. (**a**) LDF-horizontal to detect east–west orientation; (**b**) LDF-vertical to identify the north–south orientation.

The LDFs were useful to emphasize the lines in IDSM. Therefore, the output of LDFs was further processed by utilizing the majority-filter or modal-filter. The filter can be applied with user defined kernel size (e.g., 3, 5, 7, etc.) to find the most commonly occurring value (mode) for each kernel location. Additionally, the kernel shape (squared or ellipse) is also user defined. The linear features can be better approximated by ellipse shape, therefore a kernel (3*3) of ellipse shape was applied to LDF results to obtain a categorical raster data as shown in Figure 10.

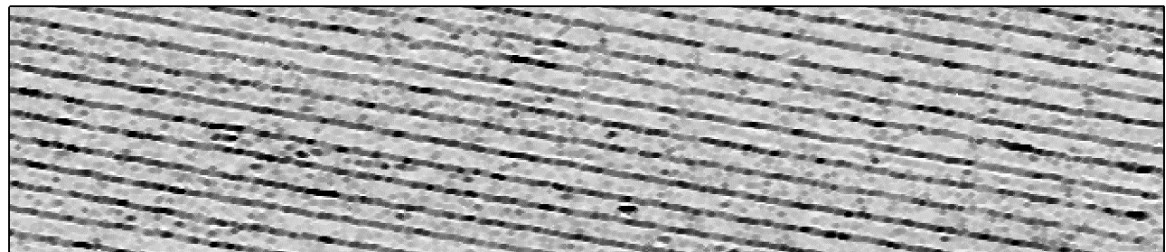

**Figure 10.** A majority- or modal-filter was used to convert LDF results (Figure 9) into a better approximation of lines and non-features.

The LDF and modal-filter based detection are visually appealing but automated extraction of detected lines is difficult, therefore, the LDF and modal-filtered images went through the image transformation process to automate the line extraction process. In order to transform the line detection results, PCA transformation was used. Initially, the PCA transformation was tested on IDSM where the results were significantly poor as IDSM consists of floating point continuous elevation values instead of categorical distinct values. However, LDF and modal-filters significantly enhance the lines features in an image as shown in Figures 9 and 10. Therefore, PCA was useful to transform LDF and modal-filter images into line and non-line feature spaces (Figure 11a). The use of PCA in a line detection application has already been well documented [65]. PCA component-1 was classified by using unsupervised K-means clustering into distinct clusters as shown in Figure 11c. The classical K-means clustering takes K points from the input image and clusters the lines and non-lines features. The K-means classification scheme is highly effective to cluster the pixel group of similar values into district clusters iteratively. Initially, a user-defined number of clusters was selected, and each cluster is refined until the desired criteria or threshold is met. The desired criteria are a number of clusters to be found in the dataset or a defined Euclidean distance. At the user end, the number of clusters along with Euclidean distance are user defined parameters. However, the direct clustering of PCA component-1 results in a very noisy output as shown in Figure 11c, therefore, the PCA component was smoothed by using a median filter to de-noise the image (Figure 11b,d). Finally, to remove the salt and pepper effect from K-means classification, a conservative smoothing filter was used for this purpose as displayed in Figure 11e. All convolutional filters and K-means clustering algorithms used are available in Whitebox GAT. Detailed documentation of used algorithms can be obtained from the user manual for Whitebox GAT software.

Whitebox GAT was developed in Java programming, therefore all the plugins and codes are available in java language. Unlike the ArcGIS model builder, Whitebox GAT does not provide the functionality of automation through the graphical language of model building. However, each Whitebox GAT tool offers easy access to underlying code through the "view code" option. In addition to this, Whitebox GAT offers the translation functionality from the native code into another programming or scripting language such as VB.NET to C# or from C# to VB.NET. Whitebox Scripter offers direct coding in Python, Groovy, and JavaScript. Therefore, codes are available to customize according to user needs. Furthermore, codes can be translated or recoded into other programming and scripting languages such as C++ and python to use in another application in order to achieve automation such as ArcGIS model builder. In this research, all the codes and tools were executed manually to process the data. As a result, the Whitebox GAT processing was carried out manually [64].

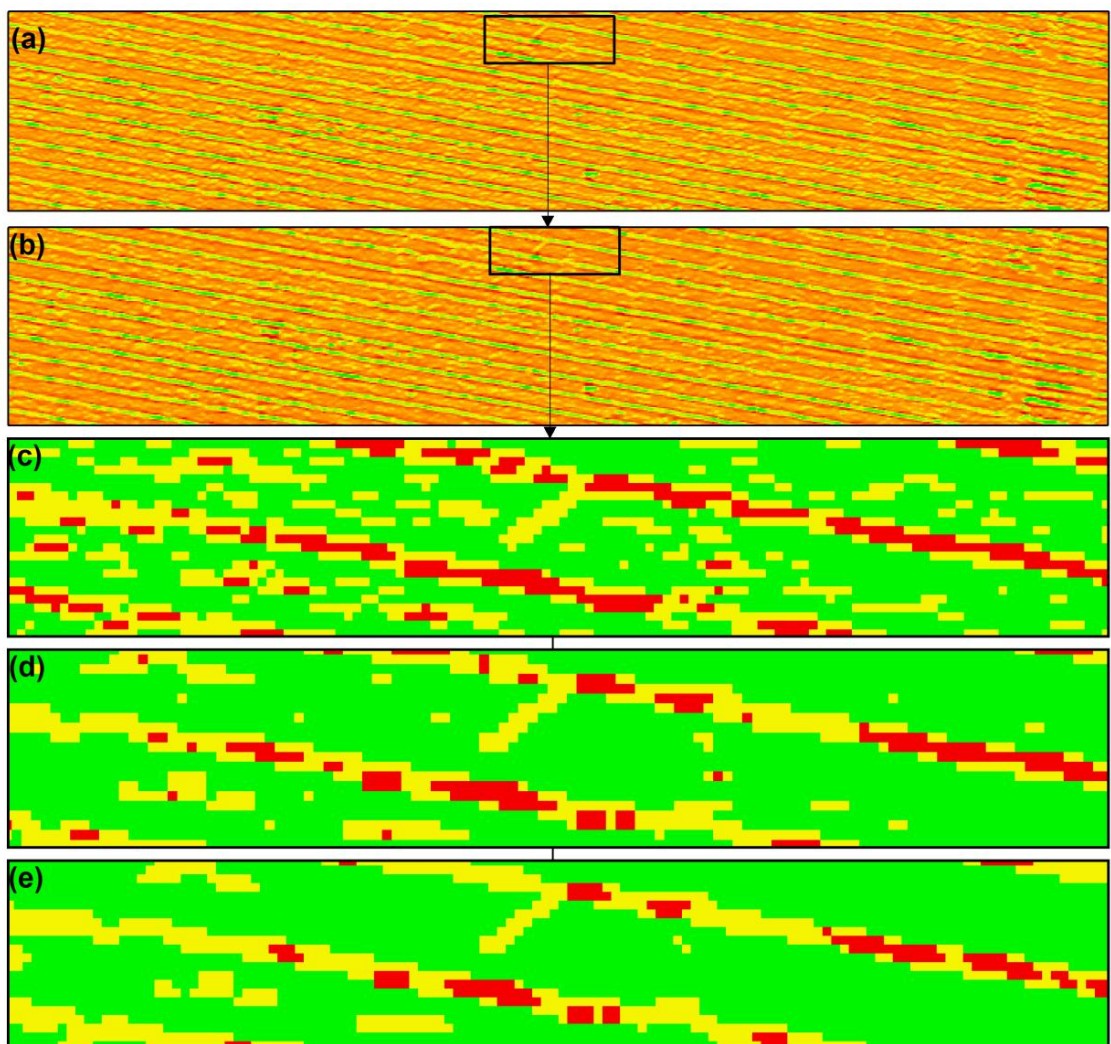

**Figure 11.** Image transformation, clustering, and smoothing process: (**a**) PCA component-1 generated by using the LDF images and modal filter images. The lines features are categorically distinct from background soil; (**b**) PCA component-1 smoothing by a median convolutional filter; (**c**) K-means clustering of PCA component-1 before applying the median convolutional filter, where background noise level is high; (**d**) K-means clustering of PCA component after median convolutional filter, where background noise is significantly reduced; (**e**) conservative smoothing filter applied on K-means clustering (d) to further reduce the background noise where line features are categorically distinct from background soil.

### 2.5. GIS-Based Vectorization Smoothing and Completion

After the convolutional smoothing of K-means classification images, the classified images were exported in ArcGIS (10.7) environment to extract the plantation rows (Figure 12a). In order to extract the plantation-rows, the images were transformed into a GIS-based vector data format. Therefore, the plantation rows were presented as vector polygons (Figure 12b). The GIS-based vectorization has the added benefit of data query, data selection, data cleaning, and data conversion from polygon to line format. The polygons that stand for plantation rows were selected by classification code and exported into a separate polygon layer as shown in Figure 12c. In terms of reducing the background noise, the raster-based smoothing was carried out in the Whitebox GAT environment through convolutional filtering. The background noises were reduced to a significant extent but were not eliminated entirely. On the other hand, the vector dataset can be cleaned by setting an area threshold to remove the non-row polygons. The background noise (non-line features) was removed by an area threshold. This results in significantly cleaned plantation rows (Figure 12d). However, for the conversion of raster

into vector datasets, the polygon shapes were generalized to remove the pixelated effect from the vector dataset. Generalization is a GIS-based process to smooth a complex polygon by removing the redundant vertices by maintaining the overall shape of the polygon (Figure 12e). Finally, the plantation rows (polygons) were converted into polygons centerlines (poly-lines) by utilizing the technique developed by Dilts, TE (2015) [66]. Generally, GIS offers the conversion from one vector data format to another vector data format such as the conversion of polygons into the poly-lines format. However, the required output was to convert plantation rows (polygons) into line features presenting the center of each polygon (Figure 12f). The centerlines extracted from row polygons were not of good quality in two different ways: firstly, the rows were not continuous line features having small gaps in extracted data; secondly, the lines were complicated by uneven shapes (Figure 12g). In order to fix the line gaps, an iterative buffering scheme of the fixed buffer size of 1.5 m was followed (Figure 12h). For a single iteration, the line gaps were fixed by 3 m by an increment of 1.5 m on each side of the gap and buffers were converted into polygons centerlines (red lines in Figure 12h). The iteration process continued until all the gaps were filled. In the final step of plantation-row extraction, the line features were generalized to obtain a smoother shape as shown in Figure 12i. The whole process was automated in the ArcGIS model builder with user defined parameters such as area threshold to remove non-row polygons and buffer size to fix the line gaps.

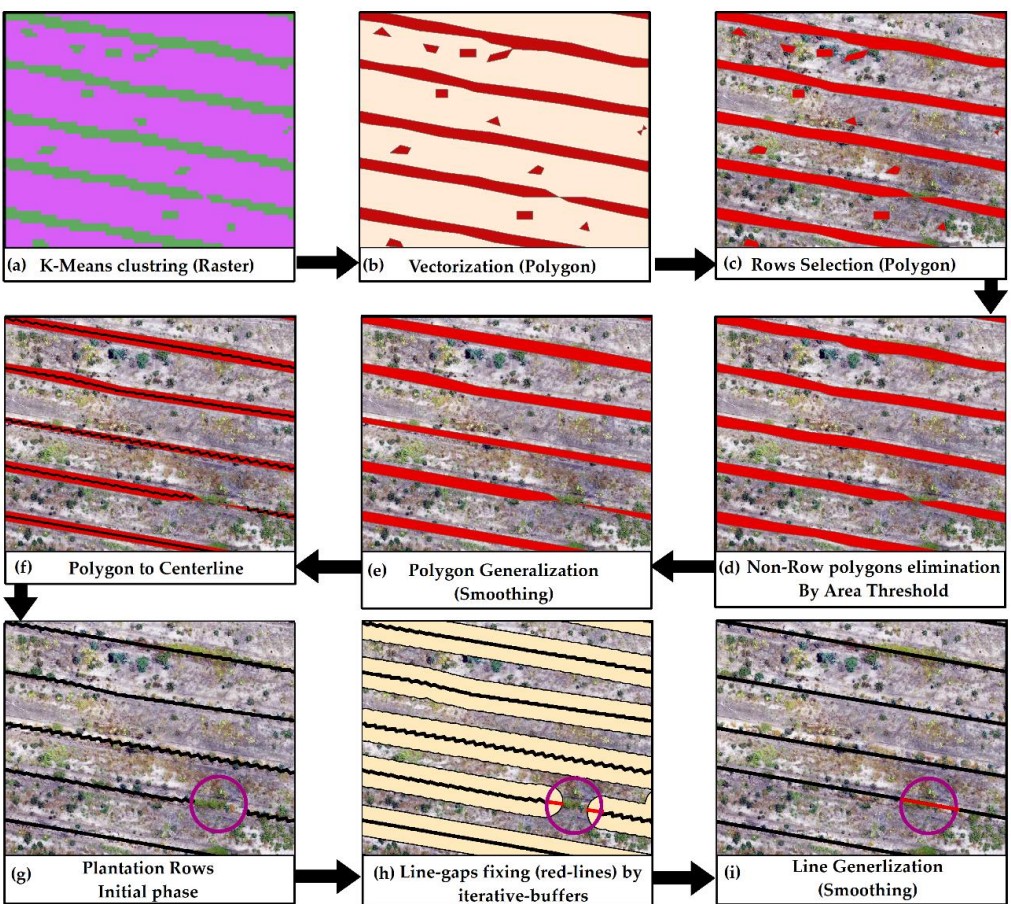

**Figure 12.** Plantation-row extraction process from raster data: (**a**) K-means classification image; (**b**) the process of vectorization to obtain polygons from pixel data; (**c**) separation of row polygons from non-row features: (**d**) elimination or non-row polygons from row polygons by an area threshold; (**e**) generalization of row polygons to smooth the shapes; (f) polygons to centerline conversion; (**g**) extracted plantation rows polylines with gaps; (h) iterative buffering to fix the gaps in plantation rows polylines; (**i**) generalization of polylines to obtain smooth and straight lines. The encircled red line segment represents the line gaps fixed by iterative buffers.

## 3. Results and Discussion

The objectives of semiautomated extraction of plantation rows from a DIPC-based DSM were twofold: first, the row extraction is based on a DSM, therefore, the line gaps represent where the plantation rows (in a DSM) were damaged and compromised; second, the line gaps were fixed to obtain the continuous plantation rows line format by utilizing the GIS-based function such as iterative buffers.

### 3.1. Plantation-Row Damage Assessment

For an ideal scenario, the plantation rows are uniformly distributed without any damage to a row surface. However, in reality, this is not always the case, the plantation rows are compromised by humans and machines during the crop sowing, field inventories, and crop harvesting, as observed in this study. A few typical examples of compromised plantation rows are shown in Figure 13. Figure 13a,b shows the locations where plantation rows were damaged and compromised respectively, however, Figure 13c shows the places where wild grass was dominant and enough 3D points were not constructed because of the homogenous surface. From the illustration in Figure 13, it is clear that such a level of damage assessment is only possible with surface-based solutions such as a DSM.

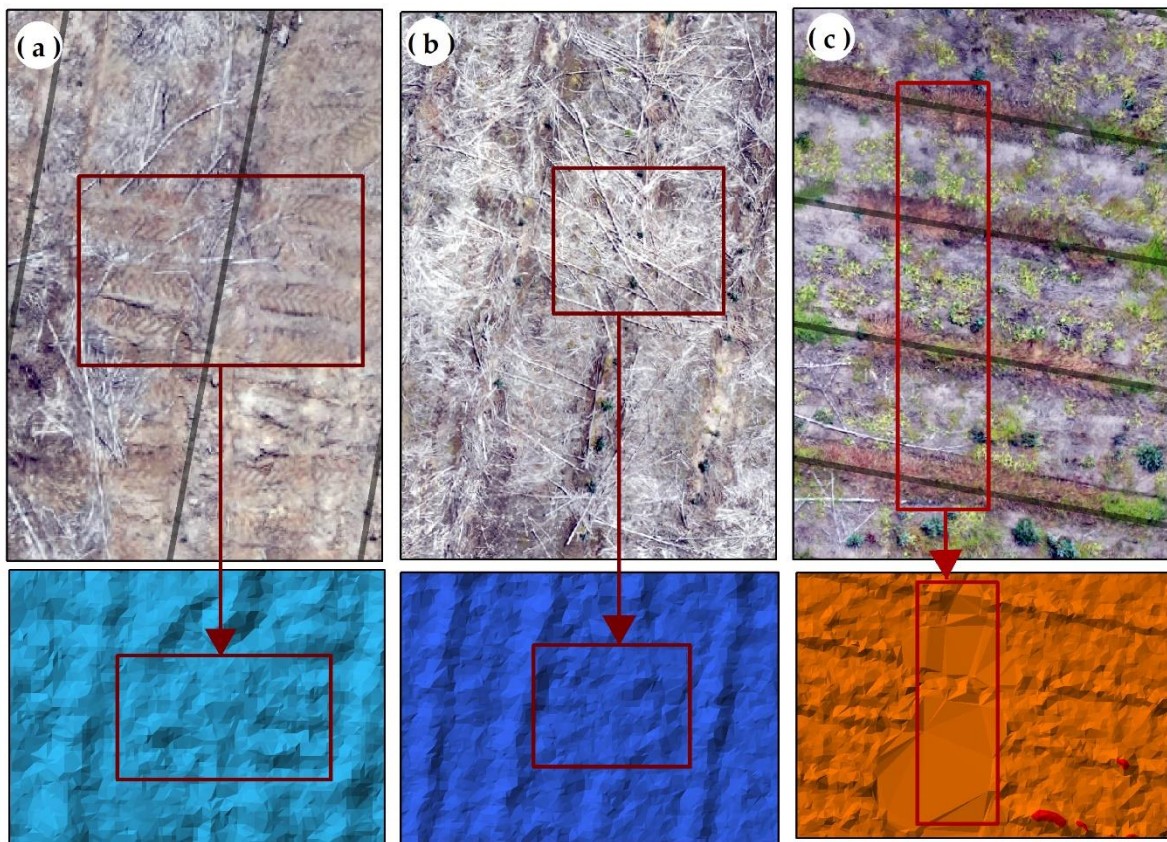

**Figure 13.** Illustration of compromised plantation rows in DIPC-based DSM: (**a**) damaged rows by machines; (**b**) The plantation rows buried under dead plantation of the previous crop; (**c**) plantation rows dominated by weeds and grass. The adequate points were not constructed by SfM software.

The surface-based plantation rows are continuous except the locations where plantation rows are damaged and compromised, as shown in Figure 13. Therefore, the extracted rows were evaluated to find the locations where rows were actually damaged. GIS-based overlay analysis was conducted to find compromised locations. In order to find the compromised locations in plantation-rows, a buffer of 3 m was placed around the extracted-rows. A 3 m buffer was chosen because the inter-row spacing was 4.5 m, therefore, the applied buffer overlaps with consecutive rows as the total distance becomes

6 m and buffers were merged into a single buffer. Due to this reason, the applied buffer covers the whole space of plantation rows along with inter-row spacing except the locations where line gaps exist. The applied method results in the location gaps where plantation rows were compromised, as illustrated in Figure 14.

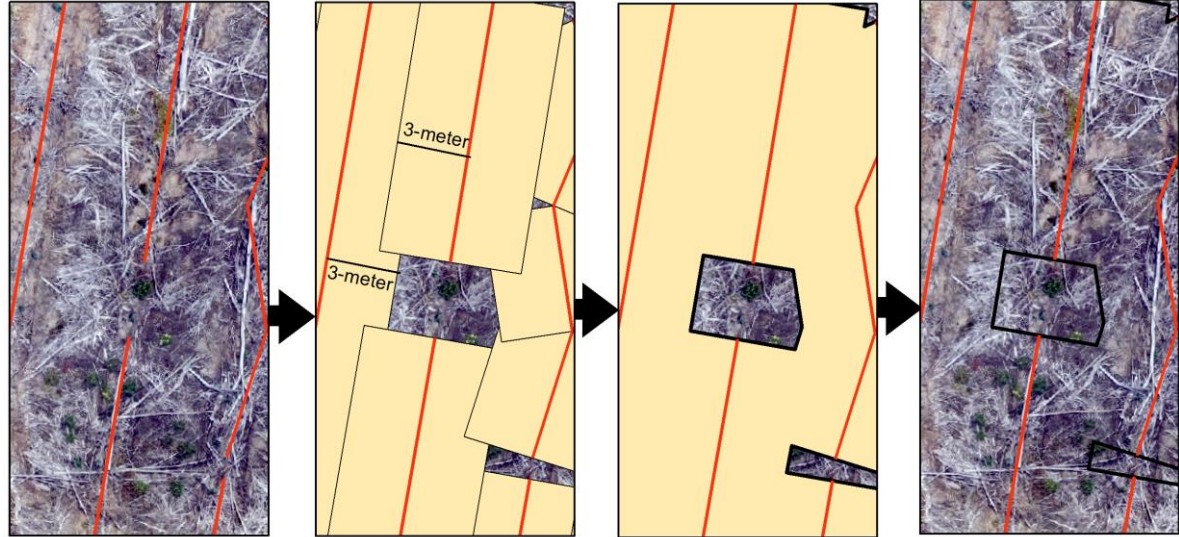

**Figure 14.** Illustration of compromised plantation-row identification. Overlapping buffers were generated to find the gaps in extracted plantation-rows. The gap identification is helpful to identify the compromised locations of plantation-rows.

By using the methodology presented in Figure 14, all the compromised locations were identified in all three compartments as shown in Figure 15.

Compartment-A was highly comprised of the background noise of dead plantation of the previous crop. Usually, crop residue is considered a management problem, however, if managed correctly can significantly improve the organic content of soil and nutrient cycling to create a favorable environment for plant growth [67]. The best management practices for PA include the proper management of crop residue. The crop residue can act as an obstruction in the drainage system thereby reducing the crop yield [68]. Compartment-A has a significant amount of crop residue, however, residue distribution is highly clustered at some locations as shown (C1-1, C1-2, C1-4, C1-5) in Figure 15a. A total of 945 sites were found in compartment-A ranging from the smallest area of 0.1 $m^2$ to the largest of 98 $m^2$. However, the residue is not affecting all the plantation-rows; therefore, identified sites can be managed and repaired in a timely manner. The second dominant factor that results in compromised plantation rows was machines used for plantation and harvesting. The damage plantation rows can be seen in Figures 13a and 15a (C1-3).

On the other hand, compartment-B was significantly cleaner having a minimum of crop residue and weeds. Therefore, the number of compromised locations (202) identified in compartment-B were fewer in number as shown in Figure 15b. Most of the locations were heavily covered with weeds and wild grass. Therefore, the locations might be apparently compromised locations where SfM 3D-scene construction was affected by the presence of a homogenous surface of grass or bright soils as shown (C2-1, C2-2, and C2-4) in Figure 15b.

In a similar fashion, compartment-C was populated by wild species of weeds, where plantation rows were significantly covered with wild grass or weeds particularly in the eastern part (Figure 15c, C3-2, and C3-3). Most of the identified locations can be attributed to SfM 3D-scene construction as *feature extraction* and *features matching* is difficult in such an environment. Therefore, the identified compromised locations might not be real.

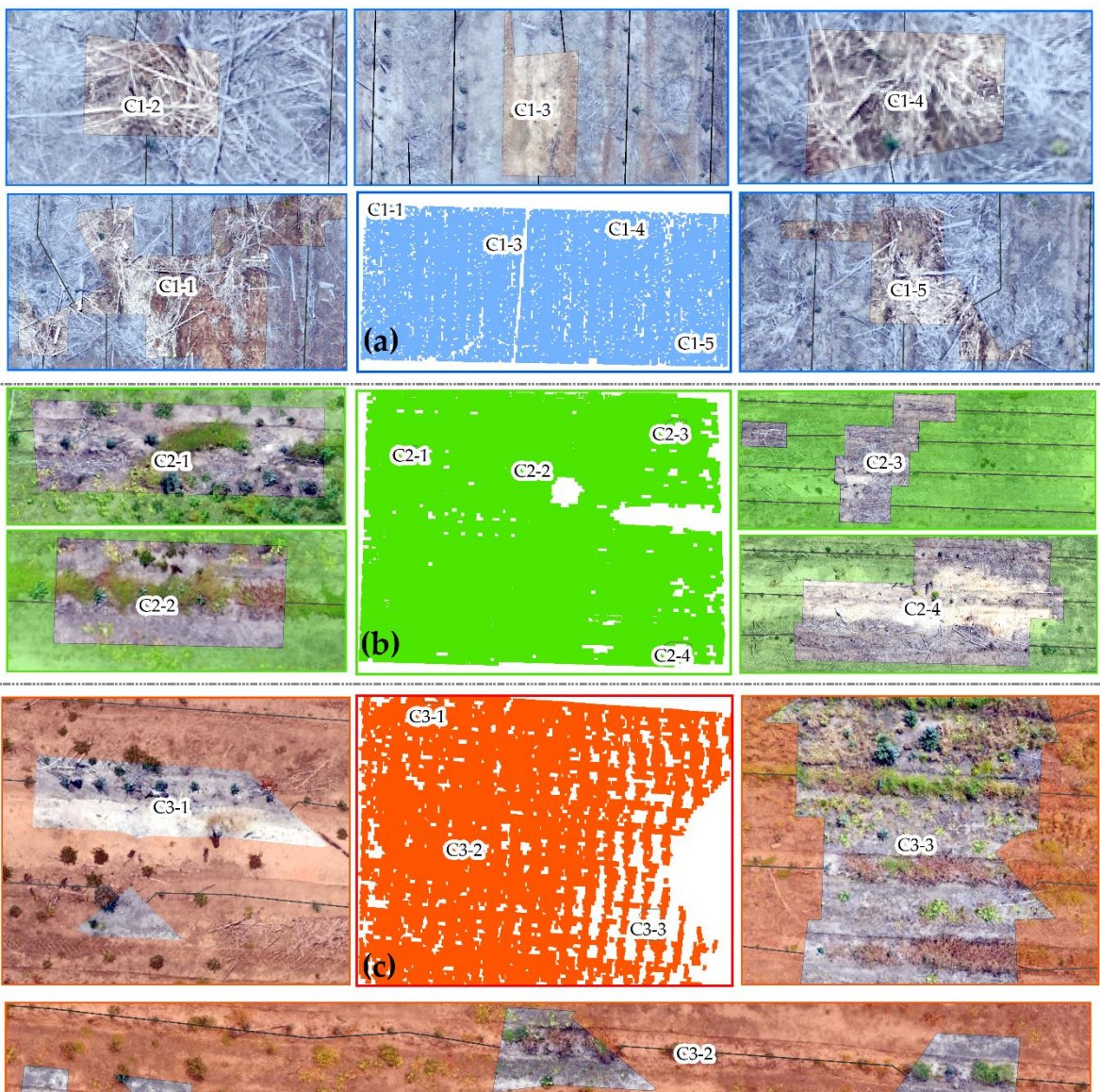

**Figure 15.** Plantation-row damage assessment for three compartments: (**a**) compartment-A, five sample locations (C1-1–C1-5) are shown; (**b**) compartment-B, sample locations (C2-1–C2-4) are shown; (**c**) compartment-C, sample locations (C3-1–C3-3) are shown.

The analysis of the compromised locations based on DSM is a very useful method in two different ways: first, all the compromised locations either real or apparent can be identified remotely; secondly, the identified location can be quantified by an area extent. This is very useful information for PA to carry out due maintenance of plantation-rows, residue management, and optimal drainage.

### 3.2. Plantation-Row Extraction Assessment

The GIS-based line gap fixing does not reach 100% accuracy, therefore, the plantation rows extraction was quantified by *correctness*, *completeness*, *quality* metrics adopted from the method developed by Heipke et al. (1997), and *F1-score* values. In order to establish an external evaluation, a reference data of plantation rows was digitized from the Orthomosaic (Figure 16). The extraction results were quantified by true positive (TP), which is the length of the extracted rows matched with reference data. The unmatched extraction results were false negative (FN), and false positive (FP) was the row segments missed by the proposed method. The values of TP, FN, and FP are shown in Table 3.

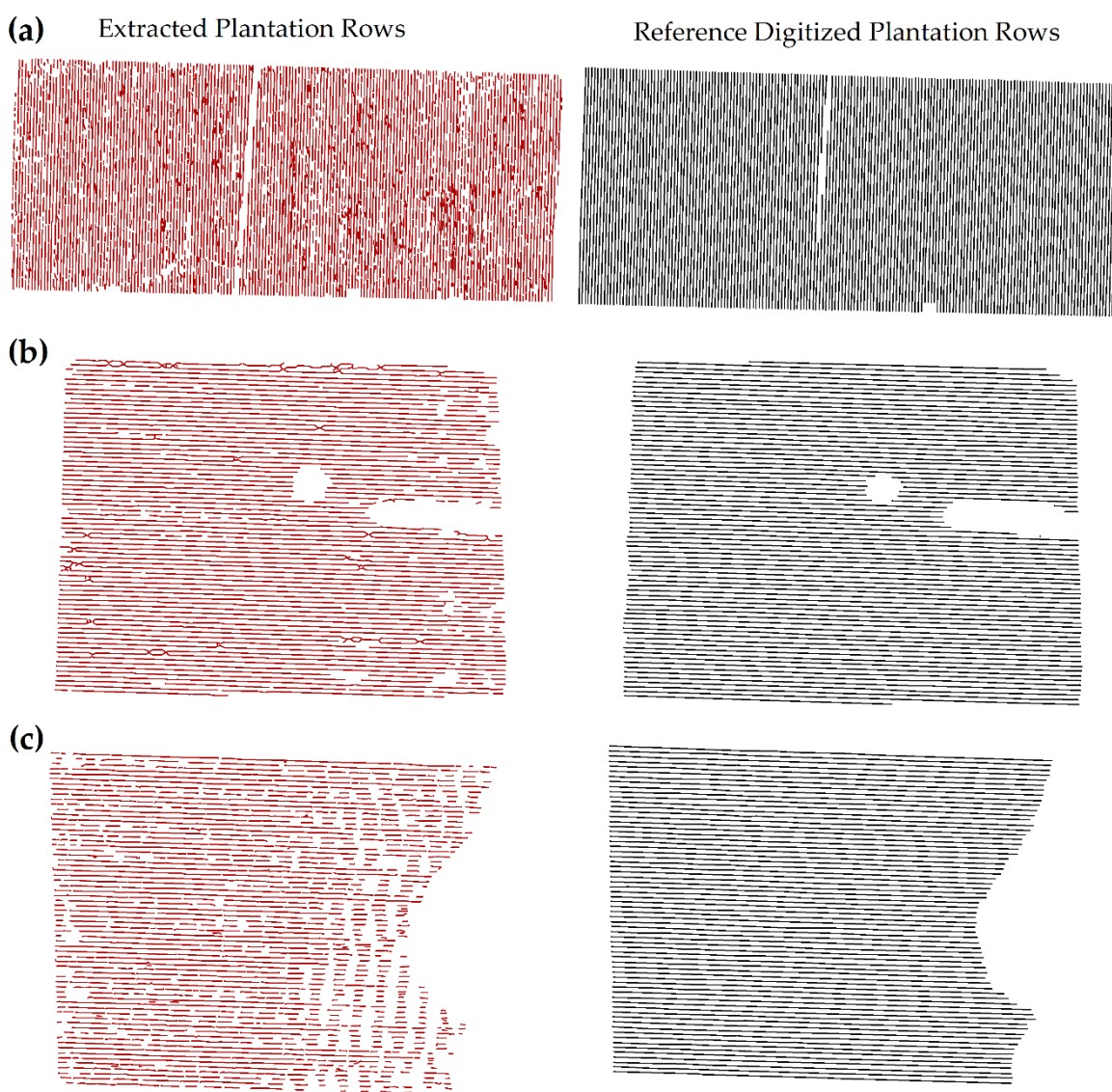

**Figure 16.** Plantation-rows: (**a**) compartment-A, plantation rows extracted from DIPC-based DSM; (**b**) row extraction for compartment-B; (**c**) extracted plantation rows for compartment-C. The grey lines represent the reference data of plantation rows digitized from the Orthomosaic.

**Table 3.** A summary of (TP), (FN), and (FP) calculated from three compartments.

|  | TP | FN | FP |
|---|---|---|---|
| Compartment-A | 54.6 | 2.54 | 6.92 |
| Compartment-B | 28.8 | 0.51 | 2.31 |
| Compartment-C | 19.60 | 0.10 | 6.11 |

Note: The values of TP, FN, and FP were calculated in kilometers.

### 3.2.1. Quantitative Evaluation

The quantitative evaluation of *completeness*, *correctness*, *quality*, and *F1-score* was done by using the following Equations (2)–(5). The respective values for each calculation are given in Table 4. The values of *completeness*, *correctness*, *quality*, and *F1-score* range from 0 (poor performance) to 1 (perfect performance).

$$Completeness \quad = TP/(TP + FP) \tag{2}$$
$$Correctness \quad = TP/(TP + FN) \tag{3}$$
$$Quality \quad = TP/(TP + FP + FN) \tag{4}$$
$$F1\text{-}Score \quad = 2 * Completeness * \\ Correctness/Completeness + Correctness \tag{5}$$

**Table 4.** Calculated values for completeness, correctness and quality by using TP, FP, and FN.

|  | Completeness | Correctness | Quality | F1-Score |
|---|---|---|---|---|
| Compartment-A | 0.88 | 0.95 | 0.85 | 0.91 |
| Compartment-B | 0.92 | 0.98 | 0.91 | 0.94 |
| Compartment-C | 0.76 | 0.99 | 0.75 | 0.85 |

In terms of *completeness*, the compartment-B plantation-row extraction was most complete as compared to compartment-A and compartment-C. This is due to the reason that the compartment-B was very clean with minimum background noises of dead plantation and weeds as already shown in Figure 2b. The plantation-row extraction was more precise with a *correctness* value of 0.98 and an *F1-score* of 0.94 with a *quality* index of 0.91. This evaluation suggests that DSM-based extraction is very successful when background noises are not very severe in an agricultural environment (Figure 16b). On the contrary, compartment-C was highly comprised of the background noise of wild species of woody weeds. Therefore, the row extraction was affected by the inability of SfM techniques to construct the 3D surface of plantation rows (Figures 13c and 16c), which result in a *completeness* value of 0.76. In terms of correctness, the extracted plantation rows for compartment-C were in complete agreement with reference data of plantation rows as the correctness value is 0.99. This evaluation also suggests that the active sensing technology such as LiDAR point clouds can significantly improve the plantation-row extractions where SfM techniques poorly perform. The overall row extraction quality was minimum with the *quality* evaluation of 0.75 and an *F1-score* value of 0.85. On the other hand, the compartment-A was also highly influenced by the background noise of the dead plantation, however, the plantation-row extraction was not highly affected by background noises as compared to compartment-C. The completeness values of 0.88 and correctness value of 0.95 suggest that the SfM 3D scene construction plays a significant part. Despite the fact that the compartment-A was highly influenced by background noises but an SfM 3D scene construction was adequate, therefore, a sufficient number of point clouds were available. However, the *correctness* value of 0.95 for compartment-A is comparatively lower than the two other compartments. This is due to the reason that compartment-A was highly affected by dead plantation of significant length and thickness oriented in random directions. Therefore, the dead plantation twigs and branches were also extracted as row features. Figures 2a and 16a clearly indicate that the inter-row spacing was overpopulated by twigs and branches of dead plants. The row extraction was based on DSM, therefore, the FN values for compartment-A were significantly higher (Table 3). The *quality* value of 0.85 and the *F1-score* value of 0.91 were lowered due to the significant contribution of FN values for compartment-A.

### 3.2.2. A Comparative Assessment and State-of-the-Art

The existing state-of-the-art was RGB-image based solutions that have been rigorously implemented, tested, and improved by successive authors in the past. Most of the existing image-based methods have been reviewed by García-Santillán et al. (2017) [69] and Ramesh et al. (2016) [70]. Firstly, RGB-image based solutions start with the process of binarization of captured images. A binary image represents the vegetated (crop rows) and non-vegetated (background soil) features. The very first drawback of the proposed solution was image quality, which can be compromised in many different ways such as the presence of shadows, weeds covering intra-row spacing, gaps between two successive or many neighboring plants of different sizes, and images acquired under different lighting conditions. The said parameters significantly affect the crop row extraction from a binary image as

binary images were compromised [69,70]. On the other hand, DSM-based solutions entirely depend on the surface elevation, therefore, are not affected by these problems. The effectiveness of using DIPC to automatically detect vineyards was tested by Comba et al. (2018) and a robust, automatic, and an unsupervised algorithm was developed [71]. However, plantation-row detection from a DIPC was never evaluated. The downside of our method is the compromised quality of elevation data. DIPC quality can be affected by homogenous surfaces of bright and dark texture, where enough matching points cannot be constructed between different images of the same scene (Figure 13c). However, the said limitation is subject to data quality not to the proposed solution. Low-cost active sensing of LiDAR technology can fill this gap in the foreseeable future [72]. Additionally, an elevation-based solution can also be underutilized in the event when crop rows and intra-row elevation differences were not captured during data acquisition and/or SfM-based processing. This conclusion was drawn from a plantation rows damage assessment where a row was flattened by a machine tire (Figure 13a). In order to fix such small gaps in plantation-row extraction, the iterative buffering solution was proposed, tested, and automated (Figure 12g–i). Secondly, most of the image-based methods were tested under controlled conditions, where vegetated patches of significant clarity with respect to background soil were present (Figure 17). In comparison to existing work, background noises were so pronounced that plantation rows were hard to identify in Orthomosaic, particularly compartment A (Figure 2a, Figure 13b). Under such conditions, RGB-image based solutions poorly perform or completely fail [70]. Thirdly, image-based solutions were tested for very small test sites comprised of a few hectares. To our best of knowledge and as literature review suggests, the tested areas did not cross 3 ha of total area [69]. Some other studies were based on test images of the same dimensions, but different quality and row extraction were carried out on an image-by-image basis [70] or region of interest basis (ROIs) [69] (Figure 17).

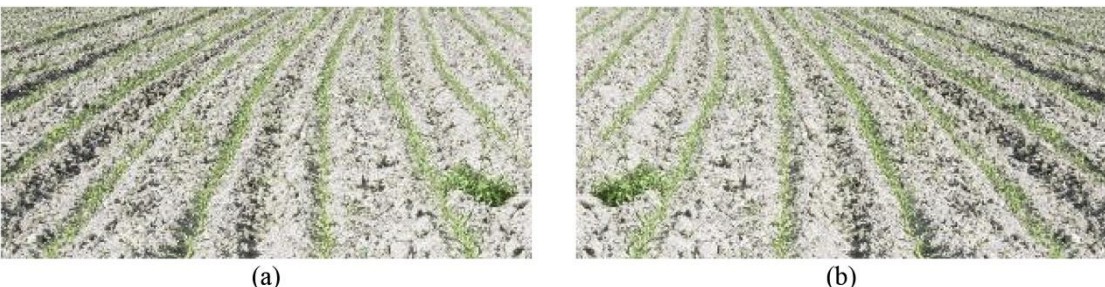

(a)　　　　　　　　　　　　　　　　　　　　　(b)

**Figure 17.** Two examples of region of interest (ROI) (a) and (b) taken from a study by García-Santillán et al. (2017). The ROIs present almost no background noise as compared to our proposed study sites (Figure 2).

Our study was tested for a vast region of 82 ha in total for three different compartments with three different levels of background noise. Additionally, the entire compartment was processed as a single image instead of a hundred of ROIs. This significantly increases the processing time, as there is no requirement of the integration of ROIs output to construct a single row from small pieces. Finally, the quantitative evaluation of RGB-image based solutions to the proposed methodology was assessed. The recent studies of García-Santillán et al. (2017) [69] and Ramesh et al. (2016) were used as they clearly quantified the row extraction results. García-Santillán et al. (2017) used a Crop Row Detection Accuracy (CRDA) term. CRDA values close to one indicate good performance. The CRDA values were around 0.92 for straight row crops in a significantly cleaner environment (Figure 17). Similarly, a study by Ramesh et al. (2016) indicates the F1-score value of 1 for good quality images. However, the accuracy starts declining to a minimum F1-score value of 0.36 when the image quality was compromised. On the contrary, in the proposed method, the F1-score values of 0.91, 0.94, and 0.85 (Table 4) for three vast compartments with significant background noise suggest the good performance of our proposed method.

The most value-added segment of the proposed research is the evaluation of compromised and damaged crop rows. The proposed solution works perfectly unless the elevated crop rows were not damaged or compromised by anthropogenic or natural means (Figure 13). Therefore, an extracted plantation-row will be broken or discontinuous if the crop row was damaged. In order to extract the seamless plantation-rows, this is not required. However, in PA this is the most valued information as compromised locations can be identified and therefore can be fixed to ensure PA (Figure 15). In comparison to the RGB-image based solution, the crop row damage assessment is not possible unless elevation information is incorporated into the crop row extraction.

Automated data processing, unsupervised feature extraction, and easier implementation of proposed methods and algorithms are very important to address a broad audience of varying degrees of expertise in programming and coding from academia and industry. The said objectives were achieved by utilizing the most widely used software applications such as ArcGIS, Agisoft, and Whitebox GAT. However, ArcGIS is a proprietary software application, QGIS (open source) could be an alternative option. One the other hand, a long list of proprietary and open source software applications are also available as an alternative to Agisoft photoscan [73]. The built-in tools and codes were easier to access and implement through automation (except Whitebox GAT). However, the mathematical framework behind every tool can be implemented through translation or coding in a single programming language or software application. Most of the tools used were unsupervised in their nature such as convolutional filters, K-means clustering, and iterative buffering. There were quite a few user-defined parameters of DSM cell size, LDFs size, and buffer size to extract plantation-rows. RGB-image based solutions on the other hand such as CNN require a large number of supervised training samples to extract plantation rows [42]. OBIA-based solutions are also supervised to define additional parameters of area, shape, and texture [74]. These parameters change with data acquisition date, time, and study sites, therefore, limited to very small study areas.

This evaluation suggests that the DSM-based solutions have proven more robust and state-of-the-art is redefined to extract plantation rows for very large areas. The proposed research will be extended to LiDAR-based solutions for different study sites having different slope variations, row orientations, inter-row spacing, different row depths, and sizes. Furthermore, this evaluation will open a new frontier for academia and industry to test, implement, and improve the proposed methods under different terrain conditions and plantation types spanning the vast regions. This research work also indicates that DIPC-based DSM obtained from UAVs images along with SfM algorithms can be used in an easier way to extract the structural parameters of agricultural land and forest plantations. The assessment was made for three plantation compartments of varying degrees of background noises of a total area of 82 ha. In PA, the use of UAVs has exponentially grown in the past few decades, therefore UAV-based DSM is not only very useful to extract the structural features such as plantation rows but also very effective to make spatio-temporal structural deformation evaluations of plantation sites. Despite the fact that plantation sites were highly comprised of background noises, the quality indexes of 0.85, 0.91, and 0.75 (Table 4) indicate that the DSM-based evaluation and plantation-row extraction is more robust and easier to implement as compared to image-based solutions.

## 4. Conclusions

This study concludes a number of significant findings that have hardly been investigated in the past. Firstly, plantation-row extraction techniques can be successfully implemented by using the 3D point clouds generated through SfM or LiDAR-based DSM for very large areas. In addition to this, the DSM-based plantation rows extraction is also useful to quantify plantation-row damage assessment in order to ensure PA. On the contrary, image-based solutions lack this potential. In a similar fashion, the present research can also be extended to LiDAR-based DSM (future directions). The research was carried out by using the most commonly available GIS and remote sensing software such as ESRI ArcGIS Desktop, Whitebox GAT (open source) and Agisoft. Therefore, complex mathematical formulation and image processing algorithms become trivial and the proposed methodology can be

implemented using open source remote sensing and GIS software without extensive programming and coding efforts. The data processing was done using SfM techniques to construct point clouds and point clouds were used to construct a DSM. These implementations are based on frequently used algorithms to generate point clouds and DSM. Furthermore, instead of using complex convolutional neural networks (CNN) and supervised training datasets to detect plantation-rows, the power of convolution techniques was incorporated by using simple convolutional filters along with unsupervised K-means clustering. Therefore, the proposed solution requires less human interaction, therefore, most of the work is automated. The existing literature is solely based on image processing and image-based solutions, and extraction results cannot overcome the limitation posed by image processing algorithms and filters. On the contrary, we refined the image processing results by GIS vector-based analysis such as fixing the background noises by an area threshold and fixing the broken rows by buffering schemes. The study was deliberately performed in three different types of background noise to assess the potential of the proposed methodology. F1-score values of 0.91, 0.94, and 0.85 are self-evident that the proposed solution gives optimum results in a poorly managed agricultural environment, except the 3D point cloud construction is not compromised.

**Author Contributions:** Conceptualization, Nadeem Fareed; Data curation, Nadeem Fareed and Khushbakhat Rehman; Formal analysis, Nadeem Fareed and Khushbakhat Rehman; Funding acquisition, Nadeem Fareed; Investigation, Nadeem Fareed; Methodology, Nadeem Fareed; Project administration, Nadeem Fareed; Resources, Nadeem Fareed; Software, Nadeem Fareed; Supervision, Nadeem Fareed; Validation, Nadeem Fareed; Visualization, Khushbakhat Rehman; Writing—original draft, Nadeem Fareed; Writing—review and editing, Nadeem Fareed.

**Funding:** This research received no external funding.

**Acknowledgments:** We are indebted to the anonymous reviewers for their careful reading of the manuscript and providing valuable suggestions, thorough insight, and recommendations to improve this piece of research work. The insight provided by four different reviewers brought significant improvement in different aspects of this research work, which otherwise was not possible.

**Conflicts of Interest:** The authors declare no conflict of interest.

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
