# Peer review of "Integration of Remote Sensing and GIS to Extract Plantation Rows from A Drone-Based Image Point Cloud Digital Surface Model"

_ijgi, doi:10.3390/ijgi9030151_

Round 1
Reviewer 1 Report
I think the article could be interesting for readers. However, it is necessary to add some essential information.
1) How was determined scale in Agisoft? Have any ground control points (GCP) been used? If so, how were they determined, where were they installed etc ..?
2) What was the calculation setup in Agisoft? In processing, it is very important to set the quality of the calculations so that the full potential of the image data can be realized. From the above figures it seems to me that the cloud on such a large area is somewhat sparse.
3) How was the point cloud checked? Since these are input data for further calculations, it would be appropriate to demonstrate the quality of the model (point cloud), for example, using control points (CP).
4) What is the row size? The rows span is given in the paper, but it would be useful for readers to specify the parameters of the line object being sought.
5) The article also states that no filtration was required. In my opinion, it would be appropriate to verify on the test data whether ground filtration would improve the speed and quality of further processing.
Author Response
1) How was the determined scale in Agisoft? Have any ground control points (GCP) been used? If so, how were they determined, where were they installed etc ..?
The overall objective of this study was to assess the potential of UAVs based DSM to extract plantation-rows. In terms of point cloud quality, we relied on the onboard GPS of the UAV system. the GPS RMSE is updated in table 1. we did not use any ground control points in this study. The UAVs images were uploaded to agisoft to construct a dense point cloud. and DIPC of point spacing of 0.2 meters was constructed.
2) What was the calculation setup in Agisoft? In processing, it is very important to set the quality of the calculations so that the full potential of the image data can be realized. From the above figures, it seems to me that the cloud in such a large area is somewhat sparse.
Agisoft is the first step to check the image quality in order to find the images with blur or bad quality. out of 1459 images, 1452 images pass the quality test. furthermore, the image density was 4-images per pixel, therefore enough matching points were available (Table 1). However, in order the properly quantify this question. we have added a new figure 3. where we have calculated area-wise point density for each compartment. we have also explained why at some location point density is not so good. a limitation of photogrammetry based point cloud solutions (homogenous surfaces or highly reflecting surface). the overall point spacing was 0.2-m. therefore, the point cloud was not sparse. For details, we refer to figure 3 and the corresponding explanation was also incorporated in the text.
3) How was the point cloud checked? Since these are input data for further calculations, it would be appropriate to demonstrate the quality of the model (point cloud), for example, using control points (CP).
The proposed methodology was subject to test the potential of a UAV based DSM. However, in future work, we will test the proposed method for high-quality LIDAR data to further extend this work. We believe that in future work, for instance, comparison with the lidar point cloud will help us to answer this question in more detail as we will have a UAV point cloud (passive sensor) and a high quality point cloud from LiDAR (active sensor). The LiDAR point clouds have very high accuracy, therefore, they will provide a standard reference to compare and contrast results with the UAV point cloud. However, the proposed methodology is designed to deal with the UAV point cloud regardless of the point cloud quality.
4) What is the row size? The rows span is given in the paper, but it would be useful for readers to specify the parameters of the line object being sought.
In Figure 4. we have augmented the row width (2 m) and height (0.3 m).
5) The article also states that no filtration was required. In my opinion, it would be appropriate to verify on the test data whether ground filtration would improve the speed and quality of further processing.
We have mentioned that "no filtration was required" because we processed the DIPC into the ground and non-ground points to get rid of background noises of grass, dead plantation and weeds. However, the algorithm-based automated extraction has resulted in very a sparse point cloud (This information is updated in Figure 4). The automated ground point extraction is subject to loss of significant numbers of plantation-row points, especially the row-tops. Therefore, the raw point cloud was chosen after experimentation. Furthermore, many authors have already proved the validity of a raw point cloud to create a DSM. We are grateful to the reviewer for asking to incorporate this vital information as it will be helpful for the others to do an evaluation of their respective sites and point clouds.

Reviewer 2 Report
Manuscript is well designed and well written. However, there are some general issues such as:
1) Inter-row spacing of 4.5 meter is very wide, only common for large trees (its okay for wider plans but not for small or slim plants)
2) Method should have been tested for at least 2 or 3 different inter-row spacing and impact of row spacing should have been addressed.
3) Depth of row has not been taken into account. Since depth of row has a direct impact on DSM, it is important to consider different depths or rows.
Minor issue: Latest development in PA mechanization, planters have GPS system. Such planters provide accurate seed/plant location, row-spacing and location.
Regardless, I think this manuscript deserves to be accepted for publication. I would let authors and editor decide if above mentioned concerns should be addressed before accepting manuscript for publication.
Minor issues:
Line 114: Objective (c) is missing
line 114: Replace, "Finally" by "finally"
Author Response
- The inter-row spacing of 4.5 meters is very wide, only common for large trees (its okay for wider plans but not for small or slim plants).
The extraction is based on DSM, if the elevation differences of rows and inter-row spacing are well-captured in DIPC or in DSM, the extraction results will not be affected. This information also included in the manuscript. However, at this point, we do not have additional or test study sites or data. The Future directions of this research to used LiDAR data as well, at that stage, we will try to obtained data with different row spacing, sizes and depth. Thank you very much for letting us know to make this an essential part of our future work.
- The method should have been tested for at least 2 or 3 different inter-row spacing and the impact of row spacing should have been addressed.
Yes, this is part of future research where the proposed methodology will be expended to LiDAR-based DSM for three different sites having different row-structural parameters of width and depth.
- The depth of the row has not been taken into account. Since the depth of the row has a direct impact on DSM, it is important to consider different depths or rows.
Yes, the Row high was also included in the text.
Minor issue: The latest development in PA mechanization, planters have a GPS system. Such planters provide accurate seed/plant location, row-spacing and location.
Autonomous solutions are still at a very early stage. Real-Time crop-row detection is also implemented and tested but such solutions are complex to carry to out and expensive for small or middle scale farmers.
Regardless, I think this manuscript deserves to be accepted for publication. I would let authors and editors decide if the above-mentioned concerns should be addressed before accepting the manuscript for publication.
Thank you very much for such valuable suggestions and recommendations. We believe that this has helped us to improve the writing part of this research and this helped us to include the non-trivial information during revision.
Reviewer 3 Report
This work focuses on detecting plantation rows from UAV images. It reports acceptable performance measures but some important components are missing here.
1. In abstract -- "The evaluation suggests that DSM-based solutions are robust as compared to RGB-image based solutions to extract plantation-rows."
I cannot find a comparison to RGB-based methods in the paper.
2. A comparison to the state-of-the-art is missing. This is an important step as the experiments were performed using the tools that do not need extensive programming and coding efforts.
3. I suggest you add a paragraph (in section 1) describing the structure of the paper.
4. Authors have mentioned the objectives of the study. But it is not clear the novelty of this work. Authors should clarify this by discussing similar works (related to objectives (a) and (b)) and comparing the proposed solution to recent literature.
5. Objective number (d) should be (c).
6. What are the components of the workflow (Fig 4) that authors have modified/improved or contributed to in addition to using standard software tools? Authors should describe them in detail comparing to the standard workflow from similar works.
7. It is not clear what components were automated in the process and what components were not. Need more details.
8. This work utilizes the inbuilt tools from standard software packages. What are the limitations of this approach?
Author Response
- In abstract -- "The evaluation suggests that DSM-based solutions are robust as compared to RGB-image based solutions to extract plantation-rows."
I cannot find a comparison to RGB-based methods in the paper.
A new section describing state-of-the art is included which describes a detail discussion with RGB-image based solutions.
- A comparison to the state-of-the-art is missing. This is an important step as the experiments were performed using the tools that do not need extensive programming and coding efforts.
A new section describing state-of-the art is included which describes a detail discussion with RGB-image based solutions.
- I suggest you add a paragraph (in section 1) describing the structure of the paper.
Thank you for this valuable suggestion, a new paragraph is added describing the structure of the paper.
- The authors have mentioned the objectives of the study. But it is not clear the novelty of this work. The authors should clarify this by discussing similar works (related to objectives (a) and (b)) and comparing the proposed solution to recent literature.
The DSM based plantation-row extraction has been accomplished in this study. There is no existing research work to make a comparison. The novelty of this work can be understood in two different ways. Firstly, instead of RGB-Image based solutions to extract plantation-row, the very first time UAVs based DSM is used. Furthermore, DSM inversion is also a novel approach which was never tried. In addition to this, simple line detection unsupervised convolutional filters were used. Secondly, most of the plantation-row extraction is based on image processing algorithms and methods. In this research work, the power of the GIS-based method is also incorporated such as automated iterative buffers to fix crop-rows gaps and area-threshold to fix background noise of non-rows polygons. In terms of comparions, the objectives (a) was compared and quantified with RGB-image based solutions. However, objective (b) is also an entirely new introduction and no such assessment exists in past research, therefore, no comparsion was possible. However, we have highlighted the significance of the proposed solution in order to ensure PA.
- Objective number (d) should be (c).
Thank you for highlight this typo mistake and it is corrected in the new version of this manuscript.
- What are the components of the workflow (Fig 4) that authors have modified/improved or contributed to in addition to using standard software tools? The authors should describe them in detail compared to the standard workflow from similar works.
The workflow implemented and shown in Fig.4 is entirely new and it was not copied or adapted from other research. The design workflow (Conceptulization) is the author's countribution which was experimented for a period of 6 months. The main idea behind this conceptualization emphasizes the fact that the DSM surface-based plantation-rows are more pronounced and simple to extract by utilizing the unsupervised line detection filter along with image transformation (PCA), and unsupervised clustering techniques.
- It is not clear what components were automated in the process and what components did not need more details.
This information also included in the revision and we have mentioned which part was not automated and why?. Furthermore, we also highlighted how to overcome this problem. In the manuscript, we have mentioned the word “semi-automated” because the workflow implemented in the Whitebox GAT environment was manual only.
- This work utilizes the inbuilt tools from standard software packages. What are the limitations of this approach?
A detail description of the usefulness of inbuilt tools is also included in the state-of-the-art section.
Reviewer 4 Report
The work presents an interesting idea. But it should be improved in some respects:
- A section describing the state of the art is missing.
- The proposed method includes key points for representation and matching of images. In this regard it is recommended to cite the following methods that adopt key points in the same way:
MANZO, Mario. KGEARSRG: Kernel Graph Embedding on Attributed Relational SIFT-Based Regions Graph. Machine Learning and Knowledge Extraction, 2019, 1.3: 962-973.
- In the experimental phase there is no comparison with methods that work on the same problem
Author Response
A section describing the state of the art is missing In the experimental phase there is no comparison with methods that work on the same problem
A section describing the state of the art is added in Results and Discussion where state-of-the-art was discussed along with proposed method.
- The proposed method includes key points for representation and matching of images. In this regard it is recommended to cite the following methods that adopt key points in the same way:
MANZO, Mario. KGEARSRG: Kernel Graph Embedding on Attributed Relational SIFT-Based Regions Graph. Machine Learning and Knowledge Extraction, 2019, 1.3: 962-973.
We are thankful for suggesting us this newly developed methods to be cited in this work. We have included the citation.
Round 2
Reviewer 1 Report
Thank you for the reactions in the cover letter. Although it is a study, it is still a scientific article and I think it is appropriate to check the calculated cloud of points. The description of your experiment should be complete, especially the data collection. The photogrammetric model can be greatly deformed, which can be very dangerous. If the photos are misaligned, the whole procedure will not work. The reader should know that the procedure of data collection you have chosen may not be correct and that the calculation must be checked. Please include a paragraph in the article to alert readers to possible errors of input data and how to eliminate them. Using the position of cameras from the on-board GPS, it is possible to partially check the calculation by means of residual errors after optimization. Please analyze your errors and include them in the article. Please add detailed Agisoft settings to the article ... calculation quality, filtration level, camera position accuracy, tie point accuracy, etc. This information will give the reader instructions on how to process input data from a specific terrain before using further analyzes and calculations. Reference [56] is not complete. For inspiration ... from the same author's team, an article about filtering was published in January 2020 in the measurement (mdpi) magazine.Author Response
Thank you very much for highlighting this very important aspect of this research work. In order to address the concerns you have mentioned in the second round or review. We have added the following information in the revised submission. I hope the said revisions have improved to make better understanding of this research work. the changes are as follows
1- Agisoft photoscan workflow (Figure 3) is added in the manuscript along with the respective literature review in order to explain the workflow.
2- New table 1. It is added to explain camera optimization and alignment results.
we hope these answers will satisfy the required changes.
Thank you very much
Reviewer 3 Report
Authors have improved the paper by addressing all the comments.
Author Response
Thank you very much for your valuable suggestions and recommendation to improve our research work.
Round 3
Reviewer 1 Report
Thank you. Now I think that It is Ok for publication.